# Ultrasound-Based Diagnostic Methods: Possible Use in Fatty Liver Disease Area

**DOI:** 10.3390/diagnostics12112822

**Published:** 2022-11-16

**Authors:** Andrej Hari

**Affiliations:** Oddelek za Bolezni Prebavil, Splošna Bolnišnica Celje, Oblakova Cesta 3, 3000 Celje, Slovenia; andrej.hari@sb-celje.si

**Keywords:** ultrasound, attenuation, predictive value, liver steatosis, non-alcoholic fatty liver disease, metabolic syndrome

## Abstract

Liver steatosis is a chronic liver disease that is becoming one of the most important global health problems, due to its direct connection with metabolic syndrome, its significant impact on patients’ socioeconomic status and frailty, and the occurrence of advanced chronic liver disease. In recent years, there has been rapid technological progress in the ultrasound-based diagnostics field that can help us to quantitatively assess liver steatosis, including continuous attenuation parameters in A and B ultrasound modes, backscatter coefficients (e.g., speed of sound) and ultrasound envelope statistic parametric imaging. The methods used in this field are widely available, have favorable time and financial profiles, and are well accepted by patients. Less is known about their reliability in defining the presence and degree of liver steatosis. Numerous study reports have shown the methods’ favorable negative and positive predictive values in comparison with reference investigations (liver biopsy and MRI). Important research has also evaluated the role of these methods in diagnosing and monitoring non-alcoholic fatty liver disease (NAFLD). Since NAFLD is becoming the dominant global cause of liver cirrhosis, and due to the close but complex interplay of liver steatosis with the coexistence of liver fibrosis, knowledge regarding NAFLD’s influence on the progression of liver fibrosis is of crucial importance. Study findings, therefore, indicate the possibility of using these same diagnostic methods to evaluate the impact of NAFLD on the patient’s liver fibrosis progression risk, metabolic risk factors, cardiovascular complications, and the occurrence of hepatocellular carcinoma. The mentioned areas are particularly important in light of the fact that most of the known chronic liver disease etiologies are increasingly intertwined with the simultaneous presence of NAFLD.

## 1. Introduction

In the last twenty years, significant progress has been made in the field of chronic liver disease (CLD) regarding the non-invasive evaluation of its main consequence, liver fibrosis (LF). With the progress of elastographic techniques, pragmatic threshold values have been defined, which enable advanced chronic liver disease (ACLD; generalized as early cirrhosis) diagnosis to be reliably defined and in a timely manner [1]. Additionally, clinically applicable norms for non-invasive elastographic assessment of the most serious ACLD complications, portal hypertension and its stages, have been proposed [1]. With the development of effective treatments in the viral hepatitis field, there has been a significant shift in the prevalence of morbidity, mortality, and the need for liver transplantation towards the fatty liver disease field. Liver steatosis (LS) is a multifactorial condition that is possibly the primary consequence of polygenic events in the organ, which escalates at the expense of various environmental influences [2]. It is currently the most common CLD worldwide, with an estimated prevalence of up to 1 billion individuals being affected [2]. Globally, the key LS causes are excessive alcohol consumption (alcoholic fatty liver disease, AFLD) and metabolic-associated liver disease, which is driven by insulin resistance and early metabolic syndrome signs (non-alcoholic fatty liver disease, NAFLD). Due to the LS pandemic, changes in the NAFLD nomenclature for metabolic-associated fatty liver disease (MAFLD) have been proposed in recent years. The NAFLD spectrum ranges from asymptomatic LS presence to parenchymal liver inflammation, termed steatohepatitis (non-alcoholic steatohepatitis, NASH), and end-stage cirrhosis and its complications of portal hypertension, liver dysfunction, and hepatocellular carcinoma (HCC) [2]. NAFLD is an independent risk factor for mortality and an important factor that affects graft survival in the liver transplantation field [2].

The evolution of elastographic techniques has contributed to the emergence of ultrasound-based applications that help in assessing and grading LS, both of which are important for prognosis and management decisions that involve CLD patients [3]. The article is an overview of non-invasive ultrasound-based applications in the LS assessment area and their potential clinical use.

## 2. LS Evaluation

LS is evaluated through liver tissue histological examination and is defined as the presence of >5% (>10% for the purpose of studies) fat cells in the liver parenchyma. As a non-invasive alternative, the use of CT examination is possible where LS causes a characteristic Hounsfield unit value decrease, but the examination is limited by poor sensitivity and by exposure to ionizing radiation [3]. In recent years, the reference non-invasive investigation for LS evaluation has, thus, become magnetic resonance examination, using the MRI-PDFF (magnetic resonance imaging proton density fat fraction) technique, which detects the presence of LS and its individual grades with great reliability [3,4]. LS is histologically divided into four clinically useful grades (S; Table 1) [2]. A clinically important entity is the transition to S ≥ 2, as this significantly increases the possibility of the presence of metabolic syndrome factors and LF progression [2]. Risk reduction related to S < 2 regression is clinically plausible and physiologically justified [4]. Due to the invasiveness of histological liver examination, limitations due to the biopsy tissue sample size (risk of LS over- or underestimation), the complexity of LS evaluation, which requires a specially trained pathologist on one side and low accessibility, and the need for radiologists and expensive examinations in the MRI-PDFF field, quantitative ultrasound-based examinations (qnUS) have been developed in recent years [3,4]. The simple use and accessibility of these methods could be particularly important, as an increasing number of CLD etiologies, and, thus, also LS etiology, are becoming multifactorial (the simultaneous presence of, e.g., NAFLD and HCV/HBV infection or NAFLD and AFLD) [5,6].

The following section is a brief overview of the qnUS methods intended for LS evaluation and their technical and clinical features. Only qnUS techniques that report the indexed LS study results are listed. Due to the rapid progress and large number of different technologies, it is very likely that some of the methods or devices are not mentioned in the article. The technical aspect of the applications is intentionally described in a short summary; the reader can refer to the proposed guidelines for a more detailed description [2]. A summation of the advantages and disadvantages of this area’s main qnUS groups is shown in Table 2.

## 3. Ultrasound Methods Used for LS Evaluation

The presence of fat cells in the liver tissue causes changes in the tissue’s physical properties, which is advantageously exploited by various ultrasound (US) imaging techniques. Liver US can be used due to the wide B-mode US availability, so as to assess the presence of LS when the steatosis degree exceeds 20% of the liver tissue. LS can be graded based on a higher liver echogenicity than that of the renal cortex, impaired visualization of the intrahepatic vessels, and impaired visualization of the diaphragm and posterior right hepatic lobe, due to US signal attenuation [3,4]. This method allows timeline LS monitoring, but it is semi-quantitative, investigator-dependent, and difficult to implement in morbid obesity cases [4].

Another semi-quantitative method, the HRI (hepatorenal index) assessment, is available on certain US apparatuses where a difference between the liver and kidney echo signal intensities can be calculated. This B-mode ratio has limited value in differentiating higher LS grades, but good performance when detecting LS [2].

Related to this field are qnUS methods, which ensure short bedside examinations, the expression of the final results in numerical form, and non-invasiveness. The methods of this field define various acoustic parameters, which are obtained from raw US data [3].

Methods that define the attenuation coefficient monitor the US signal energy loss as it travels through the tissue. The loss is greater as the liver infiltration with fatty tissue progresses. The main methods of this area are the controlled attenuation parameter (CAP) obtained with a transient elastography device and B-mode ultrasound-guided attenuation imaging [3].

### 3.1. CAP

CAP measures the ultrasound signal attenuation degree due to liver fat using a type of technology named vibration-controlled elastography implemented using a FibroScan^®^ (Echosens). The study is not time-consuming, is operator-independent, can be carried out by operators without experience in medical imaging and can be used simultaneously for liver stiffness measurement. The CAP numerical value is expressed in dB/m (decibels per meter); its values range from 100 to 400 dB/m, and the final result is the median value of 10 valid measurements (Figure 1) [4]. The device allows the use of two different probes (the M probe and the XL probe for obese patients), which have a similar diagnostic performance [3]. The examination’s weakness lies in the measurement method (the A-mode US image), which can incorrectly define the CAP value due to random measurements in the area of hepatic vessels, bile ducts, tumor lesions, or unevenly distributed liver fat [3]. Recently, FibroScan launched the Smart Exam equipment, which should improve the reliability in LS diagnosis and monitoring with a continuous CAP option. This option is meant to extend usage among severely obese patients. In addition, when the CAP measurements do not meet the predefined quality criteria, they are automatically rejected and the equipment is able to measure LS and LF at a 28% greater depth than the previous models. Study data regarding this type of application will be important for the LS field.

Examination is feasible in a high percentage of cases (>93%). The measurement is probably influenced by female gender (unclear), body mass index (BMI), and the presence of metabolic syndrome [7,8,9,10]. The age of the patient might have a similar influence [8]. Significant LS shows influence on the CAP value, especially in the presence of morbid obesity, as does mild LF. The latter was not confirmed by some comparative studies [5,9,11,12]. The influence of inflammatory activity (steatohepatitis) is probably clinically negligible, while the underlying CLD etiology influence remains questionable [5,6,13]. The mentioned influences are probably less important in the advanced LS grade (S3) [10]. 

The CAP measurement protocol has not been officially defined. Clinical use follows the LF measurement protocol specified by the manufacturer. Studies show conflicting results regarding the impact of the interquartile range (IQR) on the median CAP value reliability. A study by Myers et al. reported that the IQR showed no effect on the final outcome [12]. On the other hand, an extensive meta-analysis by Wang et al. showed that CAP validity was lower if the CAP IQR ≥ 40 dB/m [13]. This meta-analysis is supported by the latest study by Semmler et al., which shows that CAP-IQR/median values of <0.1, <0.2, and <0.3 are suitable to identify reliable measurements for diagnosing any LS (S ≥ 1) and that a CAP-IQR/median of <0.3 shows considerably higher clinical applicability, as compared with the previously suggested CAP-IQR < 40 dB/m criterion [14].

In the studies that compared the LS grade assessment against liver biopsy, it was reported that, compared to MRI-PDFF, CAP defined S2 or S3 with a moderate predictive probability (AUROC values of 0.90–0.95 for MRI-PPDF vs. 0.70–0.80 for CAP) [10,15,16]. Similar conclusions were reached in a population of patients with morbid obesity who were waiting for bariatric surgery, where the differences in the AUROC values were even slightly larger (97% for MRI-PDFF vs. 78% for CAP) [17]. MRI-PDFF is more accurate than CAP in detecting all LS grades in patients with NAFLD [10,15,16]. According to the available data, CAP is also not precise enough to delineate different LS grades, yet it might have higher diagnostic accuracy for mild LS than for moderate and severe LS [9,18].

The proposed cut-off values differ in the reported studies, but they are generally comparable if we divide them between the general population area and the NAFLD subpopulation. 

A reference meta-analysis that was based on 20 studies (approximately 3000 patients) and included various CLD etiologies defined the optimal cut-offs as 248 dB/m and 268 dB/m for S > 0 and S > 1, respectively, with very good reliability [6]. It is recommended that 10 dB/m in NAFLD/NASH patients, 10 dB/m in DM2 patients, and around 4 dB/m for each BMI unit >25 kg/m^2^ is added to the final result [6]. In a large Asian study, the researchers defined the comparable threshold values of 250 dB/m (S ≥ 1), 299 dB/m (S ≥ 2), and 327 dB/m (S = 3), respectively [5]. A study conducted in an Indian population using MRI-PPDF as a control set 262 dB/m (S ≥ 1) as an appropriate cut-off value with very good specificity and sensitivity [19].

The NAFLD subpopulation shows clear differences in the recommended cut-off values. An important study in this area identified good to very good CAP agreement with liver biopsy as a control at the following cut-off values: S ≥ 1, S ≥ 2, and S = 3 and 302 dB/m, 331 dB/m, and 337 dB/m, respectively [20]. In a similarly designed study, similar threshold values were proposed with similar predictive power [10]. A significant discordance of at least one grade between the CAP grading and the histologically assessed grade was observed in a third of the included patients, usually with S < 2 [10]. A study on a population of patients with morbid obesity and NAFLD defined the predicted cut-offs that were performed with the XL probe. The best CAP cut-offs for S ≥ 1, S ≥ 2, and S = 3 were 300 dB/m, 328 dB/m, and 344 dB/m, respectively [11].

In a viral hepatitis (HCV and HBV) study population, the threshold values for S ≥ 1 were 219 dB/m (high sensitivity and moderate specificity) and for S ≥ 2, 296 dB/m (moderate sensitivity and high specificity) with a high negative predictive value in both cases [21].

### 3.2. B-Mode qnUS Attenuation Techniques

A common feature and advantage of the investigative methods in this area is the use of B-mode US imaging, which allows direct examination of the right liver lobe and region of interest (ROI) selection within liver tissue (Figure 2). The final value is given in dB/cm/MHz (decibels per centimeter per megahertz). Part of the final result is also the measurement reliability coefficient, which varies depending on the manufacturer. The technical failure rate of these techniques seems to be low (<4%), as does the dependence of the final results on the investigator. According to the preliminary data, this is a technology that is superior to CAP and non-inferior or comparable to MRI-PDFF for identifying the presence of LS or assessing its grade [3]. The following section presents the study data of the individual US applications that are currently available on the market.

#### Individual qnUS Applications Available on the Market

iATT (Fatty Liver Attenuation Index, FUJIFILM Healthcare)

iATT was comparable to CAP in a study that assessed the presence of LS and its grade in a medium-sized patient sample. The two indicators were moderately to well correlated, even if the application protocol provided for the LF assessment was not strictly followed. The best association between LS and the results was achieved using the iATT IQR/median of <15% [22,23].

UGAP (Ultrasound-Guided Attenuation Parameter, GE Healthcare)

A comparative study of the UGAP method with an MRI-PDFF examination in a cohort with various CLD etiologies demonstrated the good ability of the UGAP method to define individual LS grades (diagnostic reliability of around 90%) [24,25]. The final UGAP value is probably influenced by the LF level [22].

TAI (Tissue Attenuation Imaging, Samsung Medison)

A study of TAI’s utility compared to MRI-PDFF showed that TAI has a good ability to detect LS and its grade in an NAFLD patient group [26].

ATI (Attenuation Imaging, Canon)

A controlled comparative study estimated that ATI’s diagnostic power was significantly higher than that of CT and CAP. The method was slightly inferior or comparable to the MRI-PDFF examination regarding the S > 2 diagnostic power assessment [27]. Similar conclusions were reached by a similarly conducted study by Ferraioli et al., who also defined individual ATI cut-off values for S > 0 in their cohort, and in a study by Spore et al., where they defined cut-off values for S ≥ 1 and S = 3 (0.79 dB/cm/mHz and 0.86 dB/cm/mHz, respectively) [28,29]. A biopsy-controlled study on phantom and healthy volunteers showed good ATI detection and quantification of moderate and severe hepatic steatosis with lesser ability to distinguish between the S0 and S1 LS stages [30].

Backscatter Coefficient (BSC)

BSC is a qnUS energy measurement, which is related to the liver tissue echogenicity during conventional US. Here, the US energy fraction that returns from the tissue is measured [2]. The method shows a good correlation with the LS grade determination in liver biopsy/MRI-PDFF-controlled studies. It depends on qualitative US data post-processing because it requires custom software [3].

A related method based on a similar principle (speed of sound) is used by the Aixplorer Mach30 apparatus. The speed of sound varies according to the fat content in liver tissue (it is low in LS cases). In a pilot study using MRI-PDFF as a reference, the speed of sound estimation exhibited very good reproducibility, good performance when diagnosing LS of any grade (S1–3), and excellent performance when diagnosing significant LS (S ≥ 2) [2].

A study comparison was also made between the LS CAP assessment and the parameters included in the Aixplorer Mach30 (sound speed plane-wave ultrasound (SSp.PLUS), attenuation plane-wave ultrasound (Att.PLUS), and viscosity plane-wave ultraSound (Vi.PLUS). The cited study defined a suitable correlation between CAP values and SSp.PLUS values. The Att.PLUS correlation with CAP values was not proven [31].

Ultrasound Envelope Statistic Parametric Imaging

This method takes advantage of the US images’ properties, which contain speckle patterns that appear in a granular form. Speckle patterns are generated by the US signal scattering due to microstructures in the tissue; therefore, speckle statistics with the backscatter envelope can describe the tissue-scattering characteristics [3]. Various statistical models have been proposed, of which acoustic structure quantification and the Nakagami distribution model have been the most widely studied.

Acoustic structure quantification is available on Canon Medical Systems devices and uses a statistical model, which, in the final calculation stage, is converted into a focal disturbance ratio (FD ratio). In fatty livers, the FD ratio theoretically decreases and the mean FD ratio can be used for LS analysis. A special feature of this technology is that the FD ratio measured in the intercostal and subcostal views does not show a significant difference and shows good agreement with the LS assessment [3].

The Nakagami distribution is a generalized statistical model that uses the Nakagami parameter to define scattering characteristics within a tissue. Early studies revealed a significant positive correlation between the Nakagami parameter and the liver tissue lipid concentrations. The method is available as tissue scatter distribution imaging (TSI, Samsung Medison), and the TSI image acquisition process is similar to that of TAI [3]. According to the preliminary study data, the method is highly reproducible, operator-independent, and easy to learn [32]. Similar conclusions were reached in related studies, which also identified a good correlation of both parameters with CAP when assessing the presence of LS [33,34]. According to the study data, its dependence on the LF grade is not clear [2].

## 4. Possible Clinical Applicability of qnUS 

The majority of the data in this area are represented by the use of CAP. The use of the remaining qUS technologies is still the subject of current research and will be important in future clinical evaluations. A summation of the possible clinical use of this area’s most important study data is shown in Table 3.

CAP can be used as a screening tool due to its negative predictive value that is sufficiently sensitive and specific at the suggested threshold values [5,9,12]. In a large Japanese study, it was demonstrated that CAP could be the most time- and cost-effective method of choice for screening large patient populations [35]. The FibroScan enables simultaneous LS and LF estimation during the same clinical session [10]. CAP is also recommended as the method of choice for the NAFLD/MAFLD patients for whom a liver biopsy will be necessary for diagnostics, as it is suitable for reliable non-invasive S ≥ S1 and advanced-stage liver fibrosis (F ≥ 3) evaluation [36]. This is particularly important in light of possible liver biopsy complications and costs and also in light of the poorer CAP performance in the S2 and S3 areas, especially in obese patients [7,10,36]. This method could also be useful for LS monitoring in patients with concurrent NAFLD and AFLD, including those with elevated BMI and DM2, where the risk of disease complications and disease-related mortality is particularly high [10,37].

In an extensive cross-sectional study on a large number of patients without CLD, it was determined that CAP-defined NAFLD is dependent on weight gain during the 1- and 10-year observation periods. Patients who significantly increased their body weight had a significantly higher risk of developing NAFLD [38]. In the cited study, it was observed that the CAP value correlated well with a mathematical model that included body mass index, neck circumference, and laboratory triglycerides and C-peptide values. The mentioned model was able to estimate the presence of S ≥ 2 with results similar to those obtained when using CAP [7]. A related study consisted of DM2 patients for whom the researchers identified an inverse CAP relationship with HDL cholesterol levels and a positive correlation with BMI and insulin levels in a multivariate analysis [39].

A study by Grgurevic et al. demonstrated that LF progression, which is a major NAFLD complication promotor, is likely to be slow. In the NAFLD and DM2 group, LF evaluation at a two-year interval is recommended [40]. CAP assessment of LS and its changes might be associated with NAFLD dynamics. In patients with NAFLD without advanced LF, high LS levels are associated with more rapid disease progression. Changes in MRI-PDFF (≥30% decline relative to baseline) are associated with NAFLD activity score improvement and LF regression and a delineated degree of cirrhosis complications. However, CAP measurement for prognostication purposes (disease progression or regression) is currently not comparable to MRI-PDFF assessment [41].

A study that evaluated the occurrence of cardiovascular complications and their association with CAP in DM2 patients demonstrated that an elevated CAP value is associated with the occurrence of both total and incident myocardial infarction or chronic kidney disease development [42]. Previous studies in this area have otherwise given contrary results, as they failed to demonstrate the association of CAP with liver-related events, non-hepatic cancer, or cardiovascular events [2,43]. It is interesting to observe that CAP-defined NAFLD shows a direct effect on increased arterial stiffness even after adjustment for conventional cardiovascular risk factors, which could be useful in observational studies for patients who need cardiovascular disease screening assessment [44]. Similar findings were identified in two studies that analyzed cardiac structures in a group of LS patients and used CAP-defined LS estimates. LS-associated metabolic syndrome was associated with left ventricular diastolic dysfunction and cardiac remodeling. These studies provide theoretical support for precise cardiovascular dysfunction prevention in patients with LS [45,46]. A slightly contradictory conclusion was reached by an observational study by Cardoso et al., who followed DM2 patients for 5.5 years and identified a protective effect of a higher CAP value on mortality, even in the cardiovascular mortality subgroup. This observation is certainly not negligible, as it points to a complex NAFLD disease spectrum that might have opposing effects on cardiovascular and mortality risk, showing a possible protective correlation with higher LS grades, at least in individuals with DM2. Part of this protective correlation might be driven by individuals’ genetic polymorphisms [47].

With nomenclature change proposals (NAFLD vs. MAFLD), the question arises as to whether the diagnostic methods from the NAFLD era would also have value in the MALFD field. A report from the NHANES III (National Health and Nutrition Examination Survey) cohort showed that the use of CAP to define LS is, as expected, remarkably comparable in both nomenclature scenarios. It also states that both designations have a similar clinical profile and long-term outcomes. However, there is an important difference, as increased liver-related mortality among NAFLD patients is driven by insulin resistance, but among MAFLD patients, it is primarily driven by coexisting AFLD [48].

An important NAFLD clinical area is the impact of weight loss interventions on LS. Unfortunately, in the last major study in this field, the researchers failed to demonstrate a significant benefit of planned weight reduction on CAP-estimated LS, despite the clear impact of lower body weight on the CAP value itself and its favorable impact on anthropometric and metabolic indicators in both the control and intervention study arms. This result could be a consequence of the current pandemic, so the authors suggest larger trials with more intensive weight-loss interventions [49]. It is important to emphasize that in the lean patients with significant LS (S ≥ 2), the survival of these patients is not influenced as much by the LS grade as it is by its consequence, LF [50].

An interesting Chinese study identified a typical negative correlation between vitamin D3 and osteocalcin values, bone metabolism metabulators, and elevated CAP values [51].

Insights from the sarcopenic obesity (simultaneous presence of obesity and reduced skeletal muscle mass) field are important because this condition is crucially related to NAFLD-related frailty and mortality [52]. The results obtained in a cross-sectional study where CAP-defined NAFLD and the LF levels were evaluated confirmed that sarcopenic obesity is independently associated with both NAFLD and the presence of clinically significant LF. The association remained statistically significant even after a multivariate analysis for socioeconomic status, lifestyle and behavioral risk factors, and metabolic conditions [53]. Regarding the influence of skeletal muscle mass on clinical NAFLD outcomes, findings have shown that myosteatosis (skeletal muscle fat infiltration) has a significant influence on LF progression [54,55,56]. Myosteatosis regression was followed by a significant decrease in LF among NAFLD patients [55]. It is not yet known whether any of the US modalities can reliably assess the skeletal muscle adiposity degree in a way that is comparable to the reference examination (CT).

Individual studies have looked at the use of the CAP, both to assess the degree of LS and to assess LS regression in patients who were bariatric surgery candidates. In a study by Kim et al., a decrease in CAP values that was relative to the surgery’s success in patients with morbid obesity was observed [57]. In the mentioned population, the use of an XL probe is recommended, and the CAP is an accurate tool for non-invasive LS grade assessment before and after the surgery itself [11,58].

With the transition of HIV infection to that of chronic disease form, research that defines the impact of NALFD in these patients is interesting, particularly because, due to the underlying disease and treatment impact, lipodystrophy is often present. A study by Lemoine et al. with MRI-PDFF as a control showed that CAP can accurately define moderate to severe LS in HIV patients and those with coexisting metabolic syndrome, and/or persistently elevated liver enzymes, and/or clinical lipodystrophy, in comparison to the general population [59]. CAP-assessed LS and the resulting associated comorbidities negatively affect patients’ quality of life and predict frailty in people living with HIV [60,61].

**Table 3 diagnostics-12-02822-t003:** Clinical applicability of presented quantitative US methods. Majority of the data are represented by the use of CAP. LS—liver steatosis. LF—liver fibrosis. NAFLD—non-alcoholic fatty liver disease. BMI—body mass index. DM2—type 2 diabetes. AFLD—alcoholic fatty liver disease. MRI-PDFF—magnetic resonance imaging proton density fat fraction. CLD—chronic liver disease. HCC—hepatocellular carcinoma. cACLD—compensated advanced chronic liver disease.

Cited Study	Possible Clinical Use
Chon YE [5] Chan W [9] Myers RP [12]	LS presence screening tool for general population Time- and cost-effective method Simultaneous LS and LF estimation
Cao Y [36] Tifan A [37]	LS/LF method of choice where a diagnostic liver biopsy is necessary Particularly important in NAFLD patients with elevated BMI, DM2 presence or dual AFLD/NAFLD etiology
Grgurević I [40,41]	LF progression prognostication in NAFLD patients Prognostication is not comparable to MRI-PDFF method
Liu K [43] Yu X [44] Peng D [45] Vitel A [46] Cardoso CRL [47]	Possible use for cardiovascular and metabolic syndrome complication prognostication Conflicting results of this area
Arora C [49] Unger LW [50] Kim KH [57] Karlas T [58]	Weight loss interventions and non-invasive LS degree evaluation
Lemoine M [59] Michel M [60] Seto W [62] Abdelaziz AO [63] Cardoso AC [64] Mak L [65] Cond D [66]	LS evaluation is possible in CLD of viral etiologies. Impact on faster LF progression in HBV and HCV patients Conflicting results in HBV area
Sugihara T [67] Izumi T [68] Oh JH [69] Mak L [70]	LS and possible increased HCC risk in HBV and HCV patients
Semmler G [71] Margini C [72] Mendoza Y [73] Scheiner B [74]	LS loss as a negative prognostic factor in NAFLD cACLD No clear LS impact on CSPH occurrence Conflicting results regarding LS impact on liver decompensation event occurrence

There are interesting data regarding the impact of LS in patients with HBV and HCV infections. The study findings by Seto et al. indicate that CAP-defined LS was an important factor that independently influenced the LF rate progression in the HBV patient group. LF risk progression was slightly lower in treated patients [62]. A study by Abdelaziz et al. in a cohort of HCV patients (genotype 4) obtained similar findings [63]. A related study showed that, even in the HCV patient population, the prevalence of LS and its association with metabolic syndrome complications is increasingly common [64]. The mentioned studies clearly show the close intertwining and mutual influence of the LS and LF process on CLD, even outside the NAFLD field. The effect of LS on LF progression in treatment-naive patients with chronic HBV infection remains unclear. A study by Mak et al., using CAP to assess LS, showed that the presence of LS is associated with a higher LF progression risk, but, paradoxically, also with an increase in the HBsAg seroclearance rate [65]. Contradictory observations were reported by Con et al., who suggest additional prospective studies in this area to clarify the relationship between LS and chronic HBV infection [65].

The CAP in combination with liver stiffness measurement (LSM) could be useful to assess the risk of HCC occurrence according to the findings of a study on a general population cohort without HBV or HCV infection and in a cohort with NAFLD and chronic HCV infection [63,67,68]. Due to the relatively widely set reference values (LSM > 5.3 kPa with any CAP and/or CAP > 248 dB/m with any LSM in the study by Sugihara et al. and LSM ≥ 8.0 kPa and CAP ≤ 221 dB/m in HCV patients and LSM ≥ 5.4 kPa and CAP ≤ 265 dB/m in NAFLD patients in the study by Izumi et al.), the question arises regarding how many patients would actually remain outside the proposed cut-off values [67,68]. Observations regarding the association between the CAP and the HCC risk in patients who were treated with a nucleos(t)ide analogue and showed suppressed hepatitis B virus replication have also been reported by Oh et al. and Mak et al. [69,70]. They suggest that routine CAP determination is important for defining the HCC risk in patients with advanced HBV-related CLD [70].

Studies in the cACLD NAFLD/NASH field show a mild, but significantly positive, CAP association with liver-related events, while the absence of CAP-estimated LS exhibited an inverse association with the same events. These results might reflect the negative prognostic LS loss meaning in this specific etiology [2]. According to the study findings, the CAP is a suitable diagnostic indicator and is comparable to histological tissue evaluation for the assessment of the absence of LS or the presence of S ≥ 2 in a population of patients with different ACLD etiologies [75]. In this subgroup, a CAP-IQR value of ≥40 dB/m did not impair CAP diagnostic accuracy [75]. The CAP measurement for ACLD patients could be significantly affected by the presence of concomitant clinically significant portal hypertension [76]. To date, however, there is no clear connection between the CAP value and the occurrence of clinically significant portal hypertension [71]. The effect of a higher CAP value on the clinical decompensation in ACLD patients is unclear. According to certain findings, a higher value is associated with an increased risk of clinical decompensation and bacterial infections independent of liver stiffness in ACLD populations [72]. On the other hand, in a study by the same study center, it was observed that the effect of higher CAP values on ACLD decompensation is probably insignificant or even protective [73]. A study by Scheiner et al. [74] reached similar conclusions regarding the first case and further instances of ACLD decompensation. A study by Mendoza et al. demonstrated a direct CAP association with the occurrence of decompensation only in patients with NAFLD-related ACLD. In patients with NAFLD/NASH, lower LS levels are probably associated with a higher LF grade and more advanced liver disease, configuring the so-called burnt-out NASH [73]. On the other hand, increased CAP values might require a different interpretation across different ACLD etiologies [73].

## 5. Author’s Opinion

qnUS technologies used for LS evaluation are extremely important when taking into account that the disease has become highly prevalent, its close connection with metabolic syndrome development, its independent impact on patients’ mortality and frailty, and the socioeconomic consequences. The advantages of the methods described in this article are often the (current) disadvantages of the field. A large number of different applications are (currently) poorly validated by studies. There are a large number of studies in small- to medium-sized populations and diagnostic value assessments based only on retrograde meta-analyses. There are also a significant number of studies that use the CAP as a clinical outcome control test, although it has not yet been fully validated. Investigations regarding LS clinical outcomes should be based on comparisons with MRI-PDFF or histological tissue analysis to the greatest extent possible.

It is of significant and positive importance that many studies within this area define the concrete role of attenuation assessment (mainly with CAP) for clinical decision making regarding NAFLD exclusion, patients with increased cardiovascular risk stratification, HCC risk assessment (especially in the viral CLD subgroup), and LS evaluation in the ACLD subgroup.

It will be important to define the impact of these non-invasive and widely available methods regarding the area of NAFLD medical interventions, such as the impact of intentional weight reduction and the treatment of metabolic syndrome factors; the importance of LS monitoring in CLD, other than NAFLD, where etiological treatment is possible; the ability to assess ACLD decompensation risk and prognostication in the coronary disease field.

Of particular note is the ability of all the listed applications to identify both LS and LF in the same clinical examination. The interplay between LS and LF in the CLD domain is complex, but this possibly shows the important influence on the progression of the portal hypertension stage in the NAFLD ACLD group.

Future challenges include the creation of study-supported pragmatic and clinically useful threshold values (CAP and other applications) that would be similar to those used in the LF field. At the same time, it is important to define how to assess LS timeline dynamics (LS progression/regression) and the impact of these dynamics on metabolic risk assessment and LF prognostication. Studies in this field should focus on hard clinical end points and multi-factorial CLDs, and, at the same time, translate the study results to the primary medical care level. Due to the rapid emergence of new applications, it is also necessary to validate how reliable these methods are according to the field’s golden standard (MRI-PDFF).

## 6. Conclusions

qnUS techniques allow us to quantitatively assess LS with moderate to very good accuracy.The advantage of these methods is the simplicity of the examinations and their non-invasiveness, as well as favorable time and financial profiles.These advantages are clinically important as LS is a major medical challenge, both in terms of the extremely high prevalence of the disease and also because of its possible serious consequences (the appearance of metabolic syndrome and progression to liver cirrhosis).With technological advancements, the presented qnUS methods could gradually take over the current golden standards in this field and facilitate early LS detection and prognostication regarding the role of LS in patients’ metabolic risk stratification and ACLD development.

## Figures and Tables

**Figure 1 diagnostics-12-02822-f001:**
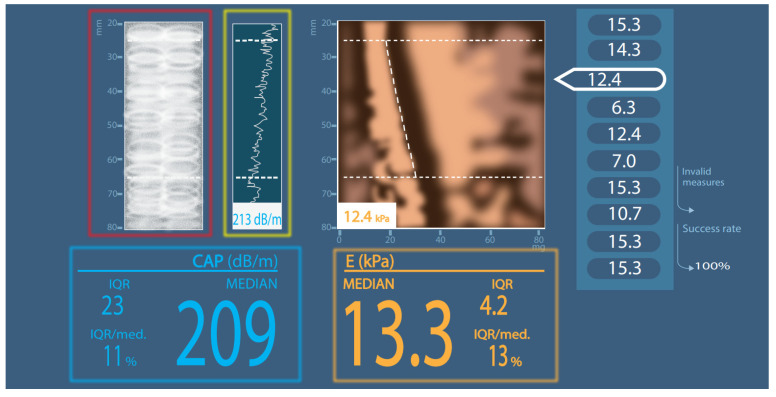
Simplified CAP measurement scheme. Red square: A-mode ultrasound picture; green square: internal liver tissue identification; blue square: CAP result; orange square: elastography result; CAP—controlled attenuation parameter; IQR—interquartile range; dB/m—decibels per meter; E—elastography; kPa—kilopascals; MEDIAN or med.—median value; mm—millimeters.

**Figure 2 diagnostics-12-02822-f002:**
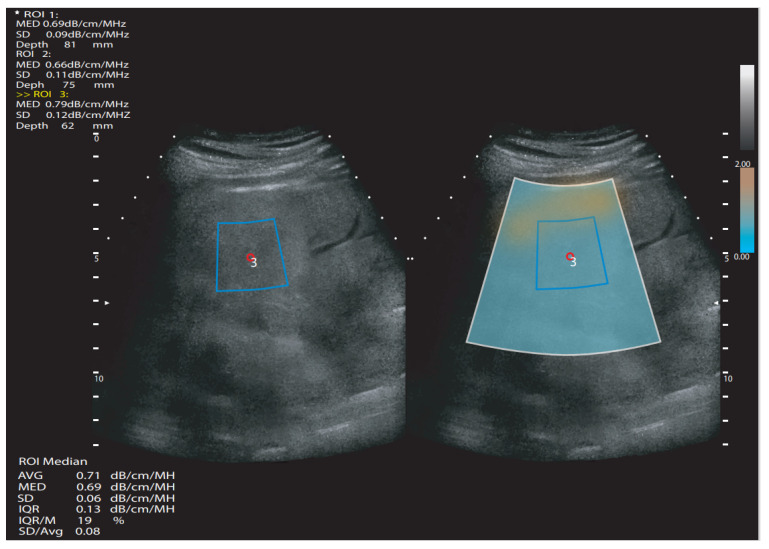
Simplified attenuation ultrasound measurement scheme. White quadrangle—area of measurement; blue quadrangle—region of interest; ROI—region of interest; SD—standard deviation. dB/cm/MHz—decibels per centimeter per megahertz; MED or M—median value; IQR—interquartile range; AVG or Avg—average value; mm—millimeters.

**Table 1 diagnostics-12-02822-t001:** Liver steatosis grades (S) according to histopathologic classification. Percentage (%) is related to the fat content within the hepatocytes.

**Grade 0**	Healthy (<5%)
**Grade 1**	Mild (5–33%)
**Grade 2**	Moderate (34–66%)
**Grade 3**	Severe (>66%)

**Table 2 diagnostics-12-02822-t002:** Advantages and disadvantages * of presented quantitative US methods. * Against reference methods (MRI-PDFF, liver histology). CAP—controlled attenuation parameter. US—ultrasound. S—liver steatosis grade. MRI-PDFF—magnetic resonance imaging proton density fat fraction. LS—liver steatosis. NALFD—non-alcoholic fatty liver disease.

	Advantages	Disadvantages
**CAP**	Easy and fast to use	Expensive equipment
	Reproducible and well accepted	Not fully clinically validated
	Concomitant liver fibrosis assessment	Modest accuraccy for S ≥ 2 and any S estimation
	Enables follow-up	Not feasible in patients with ascites
	Potential use as LS screening tool	A-mode US picture
	High NAFLD negative predictive value	Quality criteria not fully defined
**B-mode techniques**	Widely available	Expensive equipment
	Reproducible and well accepted	Scarce clinical use data
	Not inferior for S estimation compared to reference methods	Many different vendors and modalities
	Concomitant liver fibrosis assessment	Quality criteria defined by manufacturer
	Enables follow-up	
	B-mode US picture	
	Feasible in patients with ascites	

## Data Availability

Not applicable.

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
