# Peer review of "Ultrasound-Based Diagnostic Methods: Possible Use in Fatty Liver Disease Area"

_diagnostics, 2022, doi:10.3390/diagnostics12112822_

Round 1

Reviewer 1 Report

Metabolic dysfunction-associated fatty liver disease (MAFLD) is a relatively common chronic liver disease, and its incidence has a gradually increasing trend. So far, CAP and the other ultrasound-based technologies in liver steatosis are lack of uniform classification standard and threshold. This paper presents a comprehensive review of non-invasive ultrasound-based applications in the liver steatosis (LS) assessment area and their potential clinical use. I have some comments as follows.

1. The related ultrasound techniques involved in the noninvasive assessment of LS should be briefly mentioned in the part of abstract.

2. A recent meta-analysis conducted by Pu et al. (Pu K, et al. Diagnostic accuracy of controlled attenuation parameter (CAP) as a non-invasive test for steatosis in suspected non-alcoholic fatty liver disease: a systematic review and meta-analysis. BMC Gastroenterol. 2019 Apr 8;19(1):51. doi: 10.1186/s12876-019-0961-9. PMID: 30961539) showed that CAP diagnosed mild, moderate and severe LS with AUC values of 0.96 and 0.82.0.70, respectively. It seems that that the CAP technique may have a higher diagnostic accuracy for mild LS than for moderate and severe LS, which should be discussion in the part of 3.1 CAP.

3. At present, there are few literatures about B-mode qnUS attenuation techniques. It needs to be further studied to explore its diagnostic performance and characteristics. And there were studies showed that attenuation coefficient failed to distinguish between fatty infiltration of < 5% (no fatty liver) and 5% to 32% (mild fatty liver), which should be mentioned in the paper (Jesper D, et al. Ultrasound-Based Attenuation Imaging for the Non-Invasive Quantification of Liver Fat - A Pilot Study on Feasibility and Inter-Observer Variability. IEEE J Transl Eng Health Med. 2020 Jun 10;8:1800409. doi: 10.1109/JTEHM.2020.3001488. PMID: 32617199).

4. There are so many paragraphs in the paper that some of them can be merged into one paragraph.

Author Response

Dear Reviewer 1,

thank you for your thoughtful comments which enable us to improve the quality of the proposed article.

Metabolic dysfunction-associated fatty liver disease (MAFLD) is a relatively common chronic liver disease, and its incidence has a gradually increasing trend. So far, CAP and the other ultrasound-based technologies in liver steatosis are lack of uniform classification standard and threshold. This paper presents a comprehensive review of non-invasive ultrasound-based applications in the liver steatosis (LS) assessment area and their potential clinical use. I have some comments as follows.

  1. The related ultrasound techniques involved in the noninvasive assessment of LS should be briefly mentioned in the part of abstract.

Answer: We provided the requested part (see abstract).

  1. A recent meta-analysis conducted by Pu et al. (Pu K, et al. Diagnostic accuracy of controlled attenuation parameter (CAP) as a non-invasive test for steatosis in suspected non-alcoholic fatty liver disease: a systematic review and meta-analysis. BMC Gastroenterol. 2019 Apr 8;19(1):51. doi: 10.1186/s12876-019-0961-9. PMID: 30961539) showed that CAP diagnosed mild, moderate and severe LS with AUC values of 0.96 and 0.82.0.70, respectively. It seems that that the CAP technique may have a higher diagnostic accuracy for mild LS than for moderate and severe LS, which should be discussion in the part of 3.1 CAP.

Answer: Thank you for the recommended article. We included the proposed theme and citation in the 3.1. part of the article.

  1. At present, there are few literatures about B-mode qnUS attenuation techniques. It needs to be further studied to explore its diagnostic performance and characteristics.And there were studies showed that attenuation coefficient failed to distinguish between fatty infiltration of < 5% (no fatty liver) and 5% to 32% (mild fatty liver), which should be mentioned in the paper (Jesper D, et al. Ultrasound-Based Attenuation Imaging for the Non-Invasive Quantification of Liver Fat - A Pilot Study on Feasibility and Inter-Observer Variability. IEEE J Transl Eng Health Med. 2020 Jun 10;8:1800409. doi: 10.1109/JTEHM.2020.3001488. PMID: 32617199).

Answer: Once again, thank you fort he recommended articel. We included the proposed theme and citation.

  1. There are so many paragraphs in the paper that some of them can be merged into one paragraph.

Answer: We tried to approve the above mentioned part of the article.

Reviewer 2 Report

he manuscript does not contribute new knowledge. The comparison between the different methods to measure NASH  is poor

Author Response

Dear Reviewer 2,

thank you for your thoughtful comments which enable us to improve the quality of the proposed article.

he manuscript does not contribute new knowledge. The comparison between the different methods to measure NASH  is poor

Answer: Once again, we thank you for your comment. We believe that the article is a significant contribution to the discussed field since it provides the reader with a short review of the current data of the field that is currently scarcely provided in such a form on the PubMed. NASH is not the topic of the proposed review and as is well known it can not be measured non invasively yet.

Reviewer 3 Report

The paper was set as a chapter or a long narrative review article. Please consider the following:

1. Shorten the introduction and US methods . Both are too long.

2. You need at add a reference for grading fatty liver 0-3. 

3. The conclusion should be in Bullet  Points 

Author Response

Dear Reviewer 3,

thank you for your thoughtful comments which enable us to improve the quality of the proposed article.

The paper was set as a chapter or a long narrative review article. Please consider the following:

  1. Shorten the introduction and US methods . Both are too long.

Answer: We tried to improve the version as you have suggested (see the article)

  1. You need at add a reference for grading fatty liver 0-3. 

Answer: reference added (see text)

  1. The conclusion should be in Bullet  Points 

Answer: We provided the conclusion in such a form. Hopefully it complies with the rules of the Diagnostics. If not, primary form of conclusion should be preserved.

Round 2

Reviewer 2 Report

The authors improved the manuscript, however it does not add new knowledge, it is more of the same
